# A Multimodal Real-Time Feedback Platform Based on Spoken Interactions for Remote Active Learning Support

**DOI:** 10.3390/s20216337

**Published:** 2020-11-06

**Authors:** Hector Cornide-Reyes, Fabián Riquelme, Diego Monsalves, Rene Noel, Cristian Cechinel, Rodolfo Villarroel, Francisco Ponce, Roberto Munoz

**Affiliations:** 1Departamento de Ingeniería Informática y Ciencias de la Computación, Universidad de Atacama, Copiapó 1534146, Chile; 2Escuela de Ingeniería Informática, Pontificia Universidad Católica de Valparaíso, Valparaíso 2362807, Chile; rodolfo.villarroel@pucv.cl; 3Escuela de Ingeniería Informática, Universidad de Valparaíso, Valparaíso 2362905, Chile; fabian.riquelme@uv.cl (F.R.); diego.monsalves@alumnos.uv.cl (D.M.); rene.noel@uv.cl (R.N.); francisco.ponceme@sansano.usm.cl (F.P.); roberto.munoz@uv.cl (R.M.); 4Centro de Ciências, Tecnologias e Saúde, Universidade Federal de Santa Catarina, Araranguá 88906-072, Brazil; cristian.cechinel@ufsc.br

**Keywords:** multimodal learning analytics, real-time feedback, collocated collaboration analytics, social network analysis

## Abstract

While technology has helped improve process efficiency in several domains, it still has an outstanding debt to education. In this article, we introduce NAIRA, a Multimodal Learning Analytics platform that provides Real-Time Feedback to foster collaborative learning activities’ efficiency. NAIRA provides real-time visualizations for students’ verbal interactions when working in groups, allowing teachers to perform precise interventions to ensure learning activities’ correct execution. We present a case study with 24 undergraduate subjects performing a remote collaborative learning activity based on the Jigsaw learning technique within the COVID-19 pandemic context. The main goals of the study are (1) to qualitatively describe how the teacher used NAIRA’s visualizations to perform interventions and (2) to identify quantitative differences in the number and time between students’ spoken interactions among two different stages of the activity, one of them supported by NAIRA’s visualizations. The case study showed that NAIRA allowed the teacher to monitor and facilitate the learning activity’s supervised stage execution, even in a remote learning context, with students working in separate virtual classrooms with their video cameras off. The quantitative comparison of spoken interactions suggests the existence of differences in the distribution between the monitored and unmonitored stages of the activity, with a more homogeneous speaking time distribution in the NAIRA supported stage.

## 1. Introduction

Technological development in the first decades of the 21st century has grown exponentially [1]. The widespread use of these technological advances has made it possible to find alternative approaches to challenging problems in several domains, including education. One of these key challenges is managing essential skills in the 21st century [2,3,4]. Among these essential skills, also known as professional skills, is the ability to collaborate. It is important for companies and organizations to have professionals with high levels of performance in these skills since it favors their productivity in environments of great uncertainty and constant change [5,6,7]. Thus, both companies and universities are very interested in mechanisms that facilitate the adequate development of professional skills in order to precisely and objectively identify those aspects to be improved by people during their professional training or when they are already integrating teams in the workplace [3]. Active Learning methodologies are key to the development of soft skills and complex professional competencies and have been applied in different educational domains such as management, mathematics, science and engineering [8,9,10,11]. However, real-time feedback is crucial for the teacher to monitor working groups, as it is what is needed to perform precise interventions to facilitate the learning activity [12,13,14]. This issue challenges the scalability of co-located collaborative learning activities, and in the COVID-19 pandemic context, it makes it difficult to transfer active learning activities to remote learning setups.

In this context, Multimodal Learning Analytics (MMLA) offers interesting alternative solutions by integrating educational approaches with the use of technology [5]. In recent years, we have conducted exploratory studies integrating new technologies, such as MMLA and Social Networks Analytics (SNA), to study the behavior of individuals working in groups in order to solve a problem [5,15,16,17,18]. The remote academic activities have generated new challenges in the monitoring of the teaching-learning processes [19]. Teaching to “black screens” and, thus, to students that do not turn on their cameras during a class is not just a challenge when it comes to the social interaction between the teacher and the students but also to effectively assess if students are learning as well as to implement different active learning strategies [20,21]. The abovementioned reasons make the need to provide teachers with new technological resources to improve feedback evident. In this sense, the analysis of MMLA data could give the teacher feedback in real-time to make a decision to intervene (just in time) in order to ensure that working groups correctly accomplish the learning activity.

In this paper, we introduce an MMLA real-time feedback platform, namely NAIRA (meaning “eye” in the Aymara dialect). This new platform is an improved version of a previous solution based on multidirectional microphones (ReSpeaker) that we used in previous works to explore the collaborative behavior of individuals [5,15,16,17]. NAIRA is confirmed by a mobile application that captures speaking data through lavalier microphones (NAIRA APP), a cloud-based service back-end for data collection, processing, and streaming and a web application that allows the visualization of the streamed data as students interact while working in groups (NAIRA WEB). We collect audio inputs using lavalier microphones, through a custom developed mobile application (for Android and iOS) that processes and streams the vocal interaction indicators of each student to a cloud persistence platform. NAIRA WEB is a dashboard that allows the teacher to get real-time feedback. During an activity, the teacher observes the dashboard and analyzes the behavior of the groups through the following real-time visualizations: influence graph, silence time indicator, speaking time distribution pie chart, and a summary table with the total number and duration of the interactions of the group members.

The main goal of the study is to explore the effects of using the real-time feedback platform MMLA (NAIRA) on the behavior of a teacher when performing a collaborative learning activity with groups of students working remotely. To understand this in depth, we present a case study that describes the application of the NAIRA platform in an undergraduate remote class setting in the COVID-19 pandemic context. We are interested in identifying those dashboard visualizations that proved most useful to the teacher to observe student behavior, in order to analyze the way groups work and making the decision to intervene in a working group when it detects unwanted situations. We also compare spoken interaction data for both a non supervised stage of the activity and a stage supported by MMLA Real Time feedback, aiming to explore if the teacher’s interventions had an impact on the subject’s participation.

Our results suggest that the real-time feedback provided by NAIRA was highly valuable for the teacher to take pedagogical actions to facilitate the execution of the activity. The teacher could realize if the students were interacting according to the instructions, even if most of them have their cameras turned off and worked in separate virtual classrooms. The results of the quantitative analysis suggests that the participation of the students was more homogeneous in the supervised stage of the activity with respect to the distribution of the speaking time of the students.

This article continues as follows. Section 2 presents a brief bibliographic review to frame the application domain of the work. Section 3 describes the MMLA Real-Time feedback platform, the multimodal data capture process, and the components that deliver real-time feedback to the teacher. Section 4 describes the case study conducted and the design of the learning activity. In Section 5 are presented the main results of the experiments. Finally, Section 6 presents our main conclusions and future work.

## 2. Related Work

Universities have gradually started incorporating active learning methodologies in their classrooms, such as collaborative learning. However, evaluating collaboration is a challenging problem [22,23]. We need to analyze what each learner individually performs and their interactions with others while producing solutions, presenting a topic, or discussing an issue. MMLA deals with the data coming from different sources of information [24] and has grown as a prominent field of investigation, together with the rise of a new generation of emerging technologies with multimodal interfaces. The integration of data from different sources allows one to monitor interactions frequently ignored by traditional Learning Analytics strategies that normally only consider “computer-based learning contexts” [25]. Regarding the measurement and assessment of metrics in collaborative learning settings, several existing initiatives can be mentioned here.

For instance, Vrzakova et al. [26] studied collaborative problem-solving in virtual learning environments. The authors collected multimodal data from students collaboratively solving visual programming tasks using a video conferencing software. They found that task performance and the participants’ perception of collaboration were negatively correlated with idling, limited speech, and with no movement measurements. On the other hand, participants’ silence about a focus was positively correlated with performance. The authors concluded that in some cases, multimodal patterns helped to improve the predictions of task performance and highlighted that even when the predictive power is not improved by multimodal patterns, such patterns help to better contextualize and uncover details about the learning process. Moreover, Sharma et al. [27] explored how students collaborate in informal settings and how collaboration impacted students’ engagement. For that, the authors conducted a dual eye-tracking study wherein they measured the participant’s gaze, voice, learning gains, and logs while they interacted within a multi-touch game-based system. Their results showed that pairs of students with high gaze similarity (pairs who looked at the same thing at the same time) presented higher learning performances. Besides, Noel et al. [16] used multimodal data to explore the collaborative writing of userstories in software engineering. The authors collected audio recordings of students collaboratively writing user stories and derived SNA metrics from the discussions. The metrics were then linked to human-annotated information for further analysis. Among other findings, the authors highlighted that non-collaborative behavior is more observed in low productive groups with less experienced professionals and that groups with fewer interventions produced a higher number of user stories. Furthermore, Cornide-Reyes et al. [15] also used SNA techniques to analyze the collaboration of groups constructing artifacts with Lego bricks in the context of the software engineering discipline. The experiments allowed the authors to detect the important relationships and characteristics of collaborative and non-collaborative groups. Moreover, the authors reported that collaborative groups recognized the prevalence of a democratic type of leadership in the development of their work.

Research in the field of collaboration and MMLA is not restricted to experimentation and has already moved a step forward towards the development of innovative tools for collecting, measuring, and delivering processed information from collaborative settings to the stakeholders involved in the process. In this direction, Milica Vujovic et al. [28] developed an investigation focused on collaborative face-to-face learning scenarios. The authors aimed to understand to which extent a multimodal analysis based on corporal movements may help measure collaborative indicators (distance between learners, movement speed/reaction, gaze direction). For the same, the authors developed and evaluated a Motion Capture System application to help measure those indicators and consequently accelerate multimodal analysis. The authors were able to prove the effective measure and detection of the distance between learners (DBLs) and the movement speed/reaction (MS) even when the number of participants is increased. On the other hand, the authors mentioned limitations to measuring the gaze direction. The context of the experiments was a scenario wherein students with specific roles were grouped into small groups and then regrouped with other members of the group to share expertise.

In addition, Praharaj et al. [29] developed an infrastructure to automate the analysis of audio input in real-time settings. For the same, the authors considered accommodating the input from other sources in the future. The prototype uses different audio information (speaking time, number of turns, and change in loudness) mapped into a visualization that represents the growing and evolution of audio interactions. They used the metaphor of a tree that grows in different parts-tree trunk, branches, leaves, and flowers—according to the audio clues. Their system will further focus on feedback mechanisms for facilitating collaboration. Moreover, MacNeil et al. [30] developed the IneqDetect that captures collaboration in real classrooms in such a way that students can reflect on the conversations they had within their groups. The conversations are recorded by lapel microphones connected to a Raspberry pi. The tool presents visualizations of conversations in three parts: the overall talk time of the speakers, the distribution of the speaking time (measured by the Gini coefficient), and a timeline showing when each speaker spoke. The visualization helps to highlight the conversational inequality. Experiments with the tool have shown that students’ motivations and intentions to speak more have increased and they take on leadership roles. The results also showed that students were unable to correctly estimate how much they participated in group conversations. At last, Stephen et al. [31] developed and evaluated the TalkTraces system that allows on-the-fly visualizations of topics automatically detected from discussions (transcribed audio) during collaborative work. The system aims to help teams to identify the most relevant themes covered in their discussions and build a list of items covered.

The tools mentioned here are good examples of the recent advances in the field of MMLA in collaborative settings. As can be noticed, each of these tools deals with specific nuances of collaboration from a given perspective based on the collected data (audio, text, body posture, gaze, etc.). Here, we provided only a few examples of tools that support real-time feedback and those that used microphones (audio) as part of their solution. In this paper, we present a platform that provides real-time feedback on individuals’ participation in remote group activities. The feedback is visually given in the format of influence graphs representing the speech interactions of the individuals in each group, together with complementary information that allows a better understanding of the discussion dynamics within the groups. This complementary information includes silence time, effective time, level of participation (concentration of speech), and distribution of speech time. As far as we know, this is the first tool in the field that provides those functionalities in collaborative settings in real-time.

## 3. MMLA Real-Time Feedback Platform Overview

The MMLA Real-time Feedback Platform, namely NAIRA, is a cloud-based system developed to capture, store, analyze, and visualize the spoken interactions of individuals participating in remote group activities. NAIRA is made up of three main components: a data-capture mobile app (NAIRA App), a cloud-based service back-end for storage, processing, and analysis (NAIRA BE), implemented with Firebase Cloud Platforms and a web application for the configuration of learning activities, real-time monitoring of ongoing activities, and for replaying and downloading the data of already executed activities (NAIRA Web). The main components of NAIRA are shown in Figure 1 and detailed below.

### 3.1. NAIRA App

The NAIRA App is a multi-platform mobile app (Android, iOS) that allows the capturing of spoken interaction data through the mobile device microphone, or lapel microphone connected to the smartphone, for more accurate results. The app processes and sends the data of each spoken interaction event performed by the user (a group member of a collaborative activity), registering the initial and final timestamps of each interaction. The app also allows the user to input the activity for which the data will be recorded (previously defined in the NAIRA Web), specifying the group number, and to calibrate the audio input for reducing noise and setting a dynamic threshold to distinguish background noise from voice. Once the activity is configured, the calibration is performed, and the activity is started by the teacher, the app starts to stream data to the cloud services, which store, process, and send interaction indicators to the NAIRA Web. In order to convert the speech of each participant into multimodal data, two different processes were carried out depending on the operating system of the mobile device used.

NAIRA App—Android version. It was developed using the Android Studio IDE and JAVA programming language. The data collection process of the microphone devices uses the fast Fourier transform (FFT). FFT is a mathematical transformation used to convert signals between the time (or spatial) domain and the frequency domain. In this case, as the input is a periodic, limited, and discrete signal, the Fourier transform can be simplified to calculate a discrete set of complex amplitudes. The signal coming from the microphone is transformed with the AudioRecord library, which allows the management of the audio input resources for the applications and can analyze the audio directly from the hardware in real time. When the activity starts, the amplitude data is captured from the microphone every 200 milliseconds in the device (amplitudes are captured 5 times per second). This operation consists of analyzing the bits stored in the input buffer referenced by a 16-bit short variable. These bits are squared to convert it into a range of numerical data [0,32,767] where the maximum number is related to the type of the manipulated variable. If the numerical amplitude data obtained from the transform is in the range of operability established for the user, the frame is sent to the cloud. The frame structure consists of the user’s number, the second in which the capture is recorded, and the amplitude of the same capture. The data collection process continues until the activity is finished in a centralized way, calling the “stop” function of the AudioRecord library and displaying the end of the activity.NAIRA App—iOS version. It was developed for versions higher than iOS 9 using the programming languages Swift 5 and Xcode in version 11. The only difference in programming in the Android version is the real-time audio data collection process, since the AudioKit library was used to obtain the amplitude of the signal in this case. This library has a function that allows the obtainment of the signal amplitude just by setting an object of type AKFrequencyTracker with a null audio output. This object indicates the system that we are going to capture the data in real time and they will not be stored on the device. When we start the activity, the function self.tracker.amplitude is called every 200 milliseconds and it returns as amplitude a value in the range [0,1]. This amplitude value is transformed into a similar data with range [0,32,767], as the one obtained by the FFT on Android APP. If this amplitude data is in the range of operability captured for the individual, the data is sent to the Firebase database and so on in a loop until the activity is completed centrally (a change of Boolean variable in the database).

We decided to use native mobile applications to avoid compatibility problems arising from updates to hybrid solutions. The native libraries allow us to analyze the input buffer in real-time. Furthermore, they have extensive documentation and an active online community.

### 3.2. NAIRA Web

NAIRA Web allows to set up collaborative activities by defining its name, description, and an optional URL that NAIRA App will redirect to after the completion of the activity. This URL could be useful for sending web-based surveys, results, or further information about the activity. NAIRA Web also allows commanding the start of the activity, commencing the data stream from mobile devices to the cloud services, and receiving streams of processed data. The processed data stream can be visualized in four ways:1.Influence graph, which depicts the number, duration, and sequence of spoken interactions within a group (Figure 2a);2.Silent time bar, which goes from full to empty and from green to red as the time with no streams from any of the group members increases;3.Speaking time distribution pie chart (Figure 2b);4.Table with the total number and duration of the interactions of the group members as well as collaboration metrics (Figure 3).

The teacher can access a dashboard view where visualizations 1 and 2 are shown to all the groups at the same time and in real time (see Figure 4). All the visualizations have a refresh rate of 1 s. NAIRA Web also allows us to review already finished activities, allowing us to replay them within a timeline to see the changes of visualizations over time. It also allows us to view the global quantitative indicators for each group.

### 3.3. Influence Graphs, Silent Time Bar Indicators, and Speaking Time Distribution

Real-time influence graphs, silent time bar, and speaking time distribution are the main contributions of the platform within the real-time monitoring of the collaborative activity. We have detailed how they are processed in the section below.

#### 3.3.1. Influence Graph

This represents the speech interactions made by the students who participate remotely in the activity. Group interaction dynamics can be described through SNA techniques. Each group is represented as an influence graph [32] (V,E,w,f) in such a way that:The vertex set *V* is the set of participants. In this case, there are no restrictions on the number of students/participants per group.The (directed) edge set *E* represents the communicational relationships between the participants. An edge (a,b)∈E means that participant *b* is replicating interventions from participant *a*, so that *a* exerts a certain influence on *b*.w:E→N is a weight function, such that w(a,b) denotes the total number of interventions of *a* being replied by *b* during the whole activity. We denote as $w_{t}\left(a,b\right)$ the number of interventions until time *t*.f:V→N is a labeled function, such that f(a) denotes the total speaking time of participant *a* during the whole activity. We denote $f_{t}\left(a\right)$ as the speaking time of *a* until time *t*.

#### 3.3.2. Silent Time Bar

Using this component, the teacher distinguishes, graphically (color bar) or through numerical values, some aspects of the work that each remote group is developing. Specifically, the following is calculated:Silent Time: A numerical value obtained when data processing detects that the time between two input metrics exceeds five seconds. This data is not added to any node as it is considered as silence in the activity.Effective time: It represents the speaking time of the participants. It is obtained by adding up the times that are processed for the activities when more than two interventions of the same node are detected. Note that the sum of the effective times is not the total time of the activity.Level of participation: Metrics that represent the participation of a group of participants. This is based on the theory proposed by Bruffee [33] that says: when people collaborate, they need to talk to share their ideas. The more they talk, the more they sharpen their thinking skills. It follows from this theory that a group that learns best is one where there is a high level of synergy and constructive interactions among participants. This metric is obtained by relating the effective time of the participants in a group with the sum of the quiet times plus the effective time.

#### 3.3.3. Speaking Time Distribution

A complementary graphic to the influence graph is the pie chart. This graph (Figure 2b) shows the distribution with respect to the total speaking time of the group. This visualization allows the identification of the following behaviors:Dominant student: This behavior is identified when a student concentrates a high percentage of the group’s speaking time distribution. This can impair the performance of a collaborative activity. Faced with this situation, the teacher must intervene to encourage the participation of other students.Passive student: Contrary to the previous one, this is a behavior where one or more students show values far below the rest of the group members. In this situation, it is also convenient that the teacher intervenes to generate more conversation among the students.Balanced group: This is the ideal behavior in collaborative activities and results when the graph shows balanced distributions in the group’s speaking times.

### 3.4. Cloud Back-End Services

Back-end services are implemented and deployed in Firebase and node JS. Firebase is a cloud platform, integrated with Google Cloud Platform, that offers several ways to collect, process, store, and stream data. NAIRA design considers the following service:A Firebase real-time database service, for collecting data streamed by NAIRA App.A Firebase Cloud Firestore, for storing definitions of users, activities, and groups.A Node JS central processing component, for the processing of the input stream and the generation of the output values and metrics for its visualization in NAIRA Web.

At the beginning of the application development, the main providers of cloud services were: Google Cloud Platform, Microsoft Azure, and AWS Amazon. Huawei’s service was launched shortly after the implementation began and thus was not considered. We chose Firebase because the Google Cloud Platform is one of the three main cloud services providers with proven capacity in real-time storage and processing. Besides, Google has its services in Chile, being the union between the servers located in Brazil and Los Angeles, USA, which we think could provide better support in any eventuality. Furthermore, Google’s Firebase allows, only through a Gmail email ID, to use its basic services for free. In contrast, the other two services limit the time of use of their platforms unless we provide bank and company details.

## 4. Case Study

In order to explore the effect of NAIRA’s core functionalities for monitoring spoken interactions among subjects working in groups, we aimed to conduct a case study in a real-world remote learning setting. This activity allowed us to identify if the expected interaction of the subjects could be monitored and if the visualizations allowed triggering teacher interventions to improve the execution of the activity. Additionally, we explored the effect of the interventions in students’ participation and identified contextual factors that could impact the performance of NAIRA. In the following subsections, we present the case study of a Jigsaw Learning activity performed during the COVID-19 pandemic context, in a virtual classroom system. We detail the design of the learning activity (which defines the expected behavior of the subject) as well as its execution, from the perspective of the teacher monitoring the intragroup work using NAIRA Web.

### 4.1. Case Study Definition

We designed a collaborative learning activity using the Jigsaw active learning technique. Jigsaw defines an interaction structure to ensure that each student makes an essential contribution to complete a global task (the jigsaw). Jigsaw claims are to promote better learning, improve student motivation, and increase enjoyment of the learning experience [34].

This method clearly states how students must interact and when the teacher has to take action for the activity to be successful. As the activity has two stages, one with no specific guidelines about how students must interact (expert group stage) and another one that is more structured and supervised by the teacher (work group stage), we aimed to explore if the data collected by NAIRA suggested differences in the students’ participation. In consideration of the pandemic, the case study was carried out in a remote meeting setting, over Zoom.

The main research questions (RQs) of the case study were as follows:RQ1: Are the provided indicators and visualizations useful for the teacher to monitor a remote learning activity in real time?RQ2: Do NAIRA’s visualizations and data provide insights about how the remote collaborative activity was performed?RQ3: Are there differences in students’ participation between the expert group stage and the working group stage?

Regarding the second goal, it is worth noting that students did not stream from their video cameras, and while working in Zoom groups, the teacher has no access to the voice of the students unless he enters a specific group.

Considering the Jigsaw framework of reference, the case study is suitable for answering the research questions for the following reasons:Jigsaw provides a protocol that ensures that communication among the students will produce collaborative teamwork.Jigsaw defines when and how the teacher must take action to ensure the efficiency of the learning activity: the teacher must foster communication when students are not interacting and stop disruptive or dominant interactions from one or a set of participants.The real-time feedback provided by the platform allows identifying two kinds of situations. On the one hand, the silence time bar allows the teacher to identify if groups are diminishing the interactions. On the other hand, the dashboard visualizations for effective time and level of participation allows the identification of either dominant interaction (e.g., students that spoke far more time or in more instances) or disruptive interaction (e.g., a subset of students that just communicated among them).

To address the main goal of this evaluation, we collected data from four sources. The first data source was the field notes that the teacher in charge of the subject recorded during the activity. The second data source was the data and visualizations and data provided by NAIRA, previously detailed in Section 3. The third data source was the Zoom recording, which included the comments of the teacher and six observers of the research group. The fourth data source was a survey completed by the students regarding the activity design, group communication, and contribution of the group members.

### 4.2. Learning Activity Design

Following the Jigsaw design, the learning activity has the following considerations:At the start of the session, the problem is presented to all the students, specifying that each role has responsibility in acquiring specific knowledge that will contribute to solving the problem. In particular, the problem is to assess an object-oriented design from the perspective of the design principles Single Responsibility, Open-Closed, Liskov Substitution, Interface Segregation and Dependency Inversion (S.O.L.I.D.) [35].As proposed in the Jigsaw methodology, students are randomly assigned to one out of five expert groups (one for each S.O.L.I.D. principles). After that, they are again randomly gathered in Jigsaw groups of five students each, where each member is the expert in a design principle.In the expert groups stage, the students must read a definition of the principle and then analyze a design (a class diagram) to identify if that design follows the principle.In the Jigsaw groups stage, each expert must describe the principle and its assessment of the design to the other team members and solve questions. A group leader is chosen as a moderator of the activity; its main role is to ensure that all the students present their design principle and to prevent disruptive behavior (and in such a case, alert the teacher).In case the teacher is alerted by a group leader or by the group behavior (through NAIRA Web) that the activity is not working, the teacher must take action to ensure that the students behave as planned.

### 4.3. Case Study Execution

The activity considered 24 voluntary undergraduate students from an Object-Oriented Design undergraduate course. They all have previous Java programming knowledge. A brief handout of each of the activity steps and technological setup was handed two days before the activity. The students comprised five expert groups and five Jigsaw groups. We used Zoom as our remote meeting platform as it allows us to create groups and randomly assign students to them. Students were asked to connect to Zoom using their computers and also install the NAIRA App in their smartphones. The teacher had a second screen with the NAIRA Web dashboard to monitor the interaction indicators. The teacher was previously informed about the meaning of the visualizations and indicators but was free to take action based on what he believed could be an abnormal situation. Figure 5 shows the remote class setting.

According to the learning activity design, there were two stages in the activity, namely, the expert and the Jigsaw group stages. In the expert group stage, students were randomly grouped into five groups of experts (Jigsaw experts), one for each of the design principles. In the Jigsaw groups, stage five working groups (Jigsaw groups) were formed, which included a student from each expert group; all but one had five students each (the remaining group, G2, had 4 students). Both activities were monitored with the platform, however, for the analysis we considered data just from the Jigsaw working groups stage.

The video recording was processed by an ad-hoc tool to obtain the timestamped text of the comments made by the teacher and the observers during the activity, namely “the story”. The story was complemented with the teacher’s field notes, which also have the timestamp of the events noted. From all these sources, we tagged the text with the following event types: teacher’s observation, teacher’s intervention, group observers, and finished work. It is considered a teacher’s observation event in the story when the teacher is able to identify characteristics in the behavior of a particular student work group. A teacher’s intervention event occurs when the teacher enters the Zoom room of a group to perform a given action on the working group. The group observers event corresponds when the research group is able to identify characteristics in the behavior of a particular work group or student. Finally, the finished work event corresponds to the moment when the teacher identifies that the group’s work has been completed. The Jigsaw group activity had a total duration of 20 min. In Table 1, we present the number of events, its duration, and the descriptive statistical results for each type of event.

Like in most of the events related to specific Jigsaw groups, in Table 2, we detail the number and duration of events for each Jigsaw group. The distribution of the above data is shown in Figure 6.

Figure 7 shows the chronology of the events for each working group. To address RQ1, below, we provide a timeline of the most relevant events regarding both the teacher’s observations and interventions and the comments by the observers. Considering how NAIRA Web supported the real-time monitoring of the activity, the teacher and the observers agreed that there were four phases in this stage, so the events are presented and commented on according to these phases.
Phase 1—Initial coordination: This phase lasted four minutes. During this phase, the teacher allowed the students to organize themselves according to the activity instructions. In this initial stage, the teacher deliberately did not perform any observations or interventions. In the third minute of activity, both researchers and the teacher noticed a risky behavior on G2, G4, and G5 (as later shown in Figure 8) when one of the nodes increased considerably in size. As this could refer to either a student member disruptively dominating the interactions or a very active group leader, the teacher and the observers agreed to monitor the potentially disruptive subjects. It is worth noting that User 10 from Group 5 (G5-U10) engaged later with the activity and had some trouble initializing the data streaming in the already started activity.Phase 2—Development and monitoring: This phase lasted from minute 4 to minute 11 of the activity. At the beginning of this phase, the teacher made his first observations of each group based on the dashboard influence graphs shown and silent time bars for all the groups. As a result of this observation, the teacher decided to intervene in G2 and G5 because, compared to the other groups, they were very active according to the silent time bar. A very active group presents a green and completely filled up bar, which might mean there was an actual high intensity in the group’s spoken interactions or because of technical problems, such as microphone feedback, poor voice calibration, or problems in the data streaming. After visiting the individual rooms of G2 and G5, the teacher confirmed that G2 subjects were highly active, while G5 did not seem to have an intense interaction. For some reason, one student (user 10) accessed the NAIRA App with a small time lag with respect to the rest of the G5 students. This caused the bar with the group’s silence times to provide misleading feedback to the teacher regarding the initial behavior of G5. It is important to note that as time passed, the data were adjusted, and soon, the information on the silence bar was normalized.Almost at the end of this phase, the teacher noticed that for G3 (see Figure 9c), the influence graph’s node size among subjects was not balanced as compared to the rest of the groups. He complimented the observation by looking at the distribution pie chart (Figure 9a) and observed the dynamically increasing pie sections as well as the appearance of new ones, which was consistent with the Jigsaw design of the students taking turns to present their results. The use of both visualizations was very insightful for the teacher, as he could ensure that the students were performing according to the instructions, thereby preventing him from intervening unnecessarily. One of the students (G3-U0) concentrated throughout almost all the speaking time of his group (93%). However, while observing, the teacher identified that the graph changed its distribution because another student (G3-U2) began to lead the conversation (see Figure 9b).This feedback helped the teacher to confirm that the interactions were happening according to the design of the activity. It is very difficult (if not impossible) for a teacher to obtain this type of feedback without the use of these types of applications. This difficulty is maintained even if face-to-face rather than remote activities are considered.At the end of this phase, the feedback given by the platform indicated that all groups had almost balanced the influence graphs, so if students followed the work instructions, the groups should have been close to completing the activity.Phase 3—Premature closure: This phase took place during minutes 11–13 of the activity. The teacher noted that the silent time bar of G2 and G4 decreased, suggesting that interactions had ended. The teacher decided to take action; in the Zoom rooms of each group, the members reported they had finished the activity. The teacher verified which groups had finished and proceeded to give final instructions for completing the exit quiz and survey. With this, the activity monitoring for G2 and G4 was completed. Almost close to the finish this phase, the teacher noticed an increase in the silence times of the G5 group. By attending to them, he was able to confirm, as in previous cases, that this group had also completed their work. Figure 10 shows the results at the moment of finishing the work of G2, Figure 11 of G4 and Figure 12 of G5.Phase 4—Closure of the activity:This phase took place during minutes 13–20 (thus, till the end) of the activity. As in the previous groups, the teacher observed a decrease in the times of silence in the remaining groups. Therefore, he decided to intervene in G1 and G3 and confirmed that both groups had finished. The teacher repeated the closing instructions of the activities, ending the activity when 1200 s had elapsed.Figure 13 shows the results at the moment of finishing the work of G1 and Figure 14 corresponds to G3.

After finishing the activity, the subjects took a quiz on S.O.L.I.D. principles and a survey to characterize the communication during the activity, the team members’ contribution, and the activity design. The following questions were answered by all the 24 students on a five-level Likert scale.


**Group communication**
GC1. Group communication was fluid at all times.GC2. A more active role of the leader was required to improve group collaboration.GC3. Our group did not communicate well; it would have been good to receive help from the teacher to coordinate my group’s activity.

**Group members’ contribution**
MC1. The contribution of the expert on the principle of Single Responsibility was satisfactory.MC2. The expert’s contribution to the Open/Closed principle was satisfactory.MC3. The expert’s contribution to the Liskov Substitution principle was satisfactory.MC4. The expert’s contribution to the Segregation of Interfaces principle was satisfactory.MC5. The expert’s contribution to the Dependency Inversion principle was satisfactory.MC6. The knowledge achieved in the group of experts was satisfactory to me.MC7. My presentation and resolution of doubts in the group was satisfactory to my group.

**Activity design**
AD1. I consider that the time for the expert group phase was sufficient.AD2. I think the time for the Jigsaw group stage was sufficient.AD3. I think the time for the quiz was sufficient.


Table 3 shows the median and mode values of each survey question. Regarding the quiz, although the academic performance of the students is out of the scope of this article, it is worth noting that the teacher confirmed that mean grades were not different from previous years for the same topic.

## 5. Results and Discussion

### 5.1. Quantitative Results and Analysis

Using the NAIRA App and the backend services, we collected the data from the case study to examine the participation of the students both during the expert and the work group stages of the activity. The collected data are presented in Table 4. As can be seen, there are missing measurements both in expert and work group stages (n = 17 and n = 12 out of 24 subjects, respectively). These measurement problems were caused by technical problems in data collection; it is worth noting that these problems did not affect the real-time feedback to the teacher.

As commented in the definition of the study, we hypothesized that the work group stage would be more uniform than the expert group stage in terms of the participation of the subjects, considering the structure proposed in Jigsaw, as long as the teacher is able to appropriately monitor and control the activity. To evaluate this hypothesis, we compared the distribution of the participation in the expert and work group stages in terms of the speaking time and number of interactions.

Regarding RQ3, while no differences were observed for the number of interactions, the Independent-Samples Mann–Whitney U Test showed differences in distribution that, although not statistically significant (*p* = 0.097), suggest that the participation in the work group was more uniform (none of the subjects spoke more than 300 s), while in expert groups, four subjects spoke more than 300 s, which could affect the collaboration in a 12-min activity. Figure 15 shows the distribution chart and the test results of the speaking time.

It is worth noting that the four subjects that spoke more than 300 s belonged to different expert groups; thus, this could be explained as a dominant interaction that did not happen in the work group stage. Although it is not possible to state that the results are explained by the feedback provided, it is remarkable that this dominant behavior was not replicated in the working group stage. This could mean that the collaboration design of Jigsaw was effectively carried out even in a remote learning setup with just the feedback provided by NAIRA. Threats to the validity of these results are the missing measurements, but as can be seen in Table 4, the four students that spoke the most in the expert stage dimmed down their participation in the more controlled and supervised work groups stage.

### 5.2. Qualitative Analysis

Regarding RQ1 (Are the provided indicators and visualizations useful for the teacher to monitor a remote learning activity in real time?), the following observations from the case study support the idea that NAIRA successfully helped the teacher to monitor the learning activity:Of the seven interventions performed by the teacher based on real-time feedback, just one was a false-positive, due to a technical issue in data streaming. In the other six cases, valuable pedagogical actions were taken, such as confirming that a highly interactive group was working in the activity and providing closing instructions for each group.The Jigsaw activity was successfully conducted by the teacher, in a remote learning setting, through private virtual classrooms for each group based only on the feedback of the platform and the information collected in the indicators-triggered interventions.The communication among the group members and the activity design was positively evaluated by the students, and as GC3 suggests, no additional interventions by the teacher were needed. This is important considering that the teacher was guided just by the NAIRA web, and as commented, he performed precise interventions during the development and closing of the activity; additionally, the visualizations prevented him from performing unneeded—and potentially intrusive—interventions.The contribution of the team members and the design of the activity was positively perceived by the students. According to them, the use of the NAIRA app did not affect the overall perception of the activity.Considering the final grades of the students compared with a traditional activity and the teacher’s experience, it cannot be stated that the results suggest an abnormal execution of the activity.

Regarding RQ2 (Do NAIRA’s visualizations and data provide insights about how the remote collaborative activity was performed?), we claim that NAIRA allowed the teacher to see “what was going on” inside the groups, which prevented him from entering the virtual classrooms to check if the students were performing the task as requested. We support this claim through the following observations:In phase 2, supported by two complementary visualizations (the influence graph and the pie chart of interactions distribution), the teacher was able to realize that although the interactions were unbalanced in the influence graph at a given moment, the distribution of the interactions in the pie chart were consistent with the expected behavior in the Jigsaw design. In fact, the sections of the pie increased as subjects subsequently stepped in to report their findings.As each group ended the activity at its own time, the silent time bar successfully represented the lowering in the spoken interactions, thus allowing the teacher to timely assist (and almost to anticipate) the group in performing the final actions to close the activity.

Although these results were strongly case dependent, they show the feasibility of performing a learning activity based on the real-time feedback provided by the platform. Moreover, it is interesting to compare how real-time feedback triggered behaviors that were very close to a co-located physical setting:In phase 1, when two groups seemed to have dominant interactions with the presence of a dominant student, the teacher was triggered to track if this behavior persisted. This behavior was comparable to keeping track of a student speaking out loud in the classroom and checking if they were effectively working as requested.Similarly, in phase 3, the teacher visited highly interactive groups, which were the balanced groups; this is similar to checking if bustling groups are being enthusiastic about the work or are not working as requested.Even when technical problems were detected, it was comparable to when a student engaged late into a group, introducing variability in the expected behavior and, thus, forcing the teacher to provide special instructions to carry out the activity.

We think the above-mentioned technical issue could happen for two reasons. Firstly, because of a poorly performed speech calibration in the NAIRA App (see Section 3), which could cause some student interactions to not be recorded on the platform. Secondly, because of ambient noise in the student location. In the latter case, the problem could appear if this noise has a wavelength very close to the one registered in the NAIRA App in the student’s microphone calibration process. This issue would cause a permanent capture of the microphone that does not distinguish between the student’s voice and the ambient noise. To our knowledge, and according to the data, no other issues regarding noise or poor calibration affected the activity.

One of the differentiating features offered by the NAIRA platform is the possibility of making a post-analysis of the behavior of the groups. This is possible thanks to time stamps that allow a dynamic interaction with each visualization. To qualitatively analyze the behavior of the groups in Figure 16, Figure 17, Figure 18, Figure 19 and Figure 20, it is possible to observe the distribution in the speaking times of each group.

Figure 16 and Figure 17, groups G3 and G5 respectively, reflect the existence of a dominant student in each group. As discussed in RQ2, the teacher intervened in each group by observing this undesirable behavior for collaborative learning activities. After the intervention, the teacher could verify that the behaviors were caused by the environmental conditions of the students involved (User0 of G3 and User10 of G5).

Figure 18 and Figure 19, groups G1 and G2 respectively, show a very similar behavior in time. Both groups present, in the beginning, a dominant student that groups more than 50% of the speaking times in each groups. From an intervention, the teacher could verify that this initial dominance was caused as it was a student that led and organized the development of the activity of each group. However, in Figure 18b and Figure 19b, it was already possible to evidence changes in the pie charts a product of a greater participation of the other students in the group. In the Figure 18c and Figure 19c, we can observe the distribution that each group has at the end of the activity. Here, it is already possible to observe a better balance between students.

Finally, Figure 20 shows the evolution in the distribution of G4 speech times. The pie chart reflects a Balanced group behavior. This group meets the desired and expected characteristics when performing a collaborative learning activity. As an additional feature, G4 was the first to complete the activities and the knowledge assessment yielded positive results for each member of the group. These findings allow us to pose future research questions to explore the relationships between a group’s collaborative behavior, time spent performing tasks, and knowledge assessment results.

## 6. Conclusions and Future Work

Real-time feedback for monitoring activities performed remotely is becoming more relevant. The use of sensors and the use of MMLA techniques will be key in the generation of quality information for real-time feedback. While it is possible to identify tools that generate speech time visualizations in the literature, to our knowledge, NAIRA is the first platform in the field that provides such functionality in real-time collaborative environments.

The main goal of the case study was to explore the effects on the behavior of a teacher that uses a real-time feedback platform MMLA (NAIRA) in performing a collaborative learning activity with groups of students working remotely. Regarding RQ1, our results suggest that the silent bar and collaboration visualizations (influence graph) trigger meaningful interventions of the teacher. Of seven interventions, all but one yield relevant actions in the context of the learning activity, allowing the teacher to identify interactions decreases in teams that finished the assignment and to supervise high activity groups. In addition, as the dynamic participation ratios of each student related to the interaction design of the learning activity, the teacher could figure out that teams were working according to the instructions, thus preventing him from performing unnecessary interventions.

Regarding the second case study goal, as well as minor technical issues, we identified context factors that can impact the quantitative monitoring of learning activities. These context factors are comparable to those that can affect collocated learning setups. For instance, students that engage in the activity later than the rest of the group members introduce anomalies in the group behavior, leading to erratic data capture and affecting the teachers’ ability to monitor the activity. In the same fashion, technical issues with microphones or ambient noises introduce bias to the spoken interaction data collection, thus requiring the teacher to observe the students’ interactions more carefully or even take a different monitoring strategy. These technical issues are also comparable to a physically collocated activity where the teacher must reach out to groups that cannot hear or see because of physical interruptions such as noises or space constraints in the classroom.

Although exploratory, quantitative results suggest that the more uniform participation of the students in the work groups stage, which was predicted by the Jigsaw design, was successfully achieved. Considering the setting and NAIRA as the only monitoring tool for the teacher, this is an interesting result for inspiring further research on the effect of MMLA real-time feedback for the effective management of students’ participation under remote learning settings.

Future work, aiming to explore the effect in both the performance of each student as a team member and in the overall group interaction, will be focused on exploring real-time feedback visualization in working groups. In addition, technical improvements will be developed for noise reduction and re-synchronization of team members that engage late in the activity. Finally, we have considered a research on the efficiency of the NAIRA App and NAIRA Web to analyze issues of scalability of the platform.

Finally, this experience allowed us to demonstrate the importance of the use of technologies to support academic activities today, especially when the COVID-19 pandemic forced us to carry out a large part of our activities remotely. Undoubtedly, the platform presented in this article provided the teacher with a powerful feedback tool with which it was possible to observe working behaviors in remote groups, intervene when he detected an irregular situation in the work of the groups and, finally, detect possible problems in the development of the activity.

## Figures and Tables

**Figure 1 sensors-20-06337-f001:**
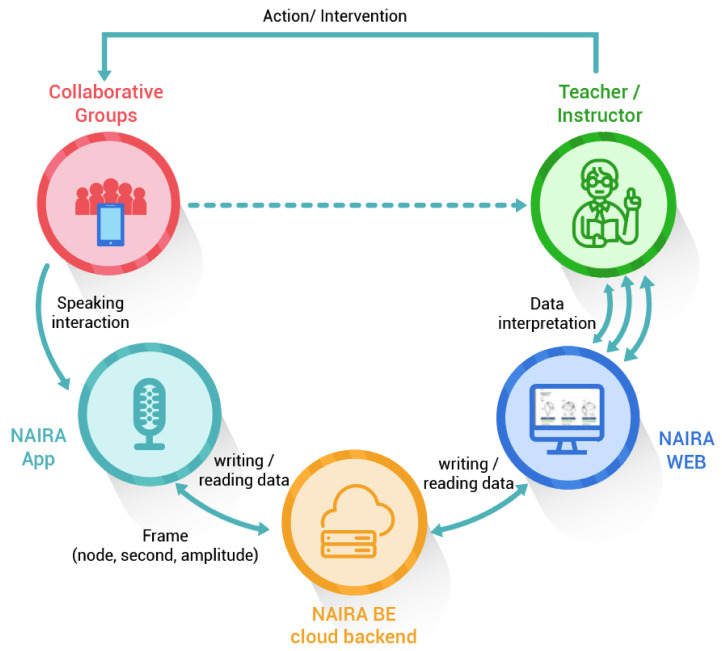
General diagram of the proposed solution.

**Figure 2 sensors-20-06337-f002:**
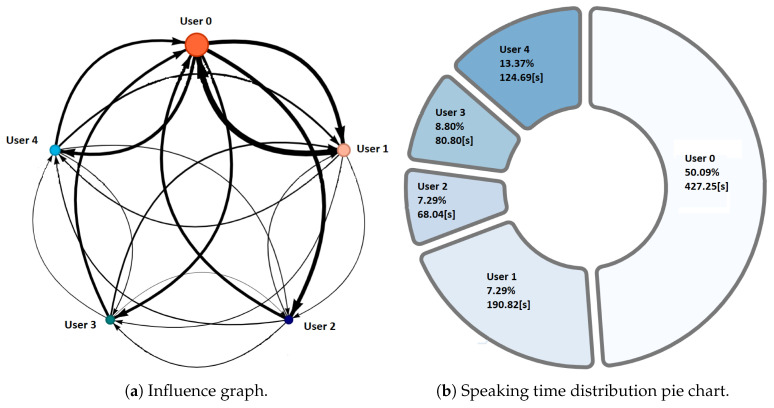
Some visualizations available in NAIRA Web for real time feedback.

**Figure 3 sensors-20-06337-f003:**
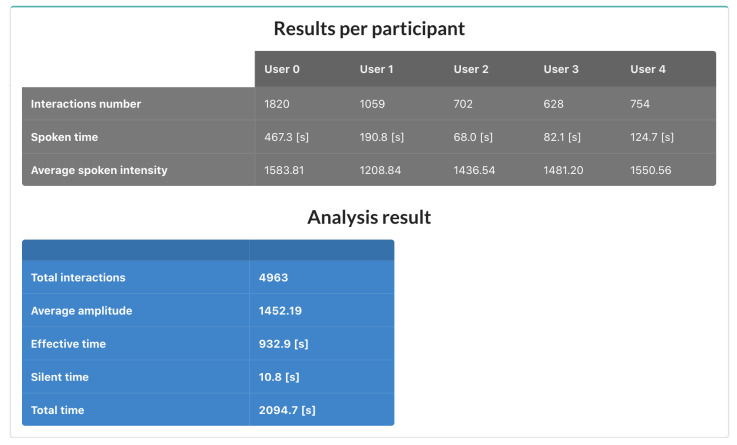
Summary of analyzed data of NAIRA.

**Figure 4 sensors-20-06337-f004:**
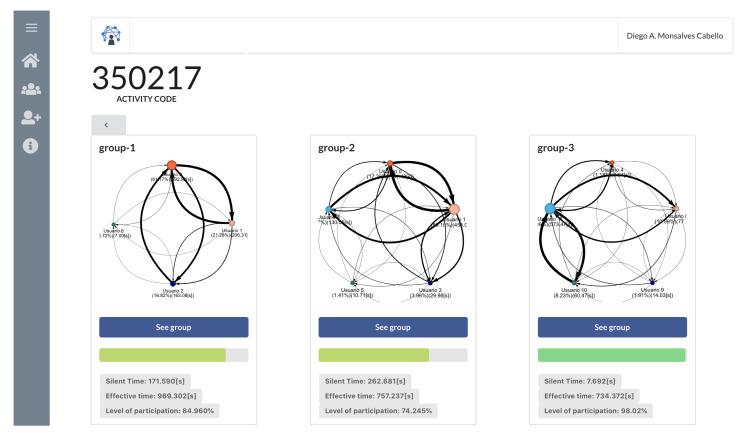
Dashboard view of influence graph and silent time bar for multiple groups.

**Figure 5 sensors-20-06337-f005:**
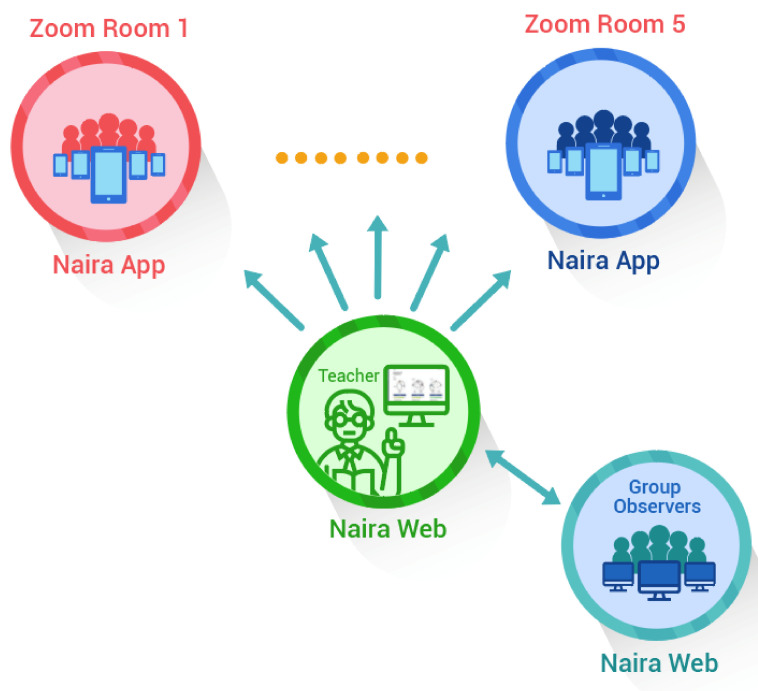
Remote class setting.

**Figure 6 sensors-20-06337-f006:**
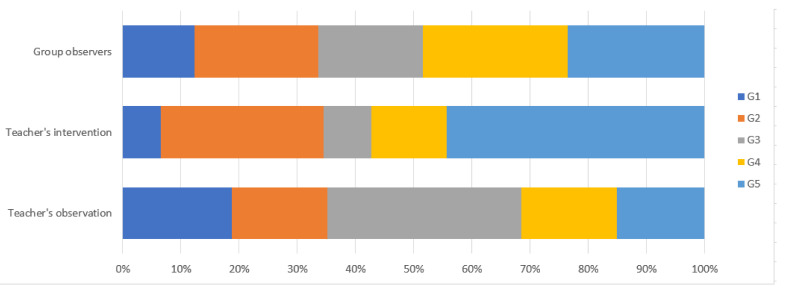
Quantitative summary of events identified by each working group.

**Figure 7 sensors-20-06337-f007:**
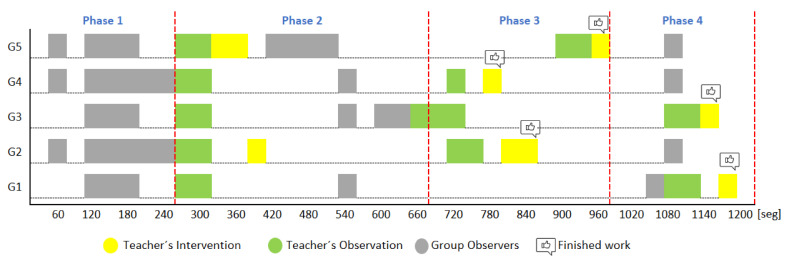
Chronology of events identified during the activity.

**Figure 8 sensors-20-06337-f008:**
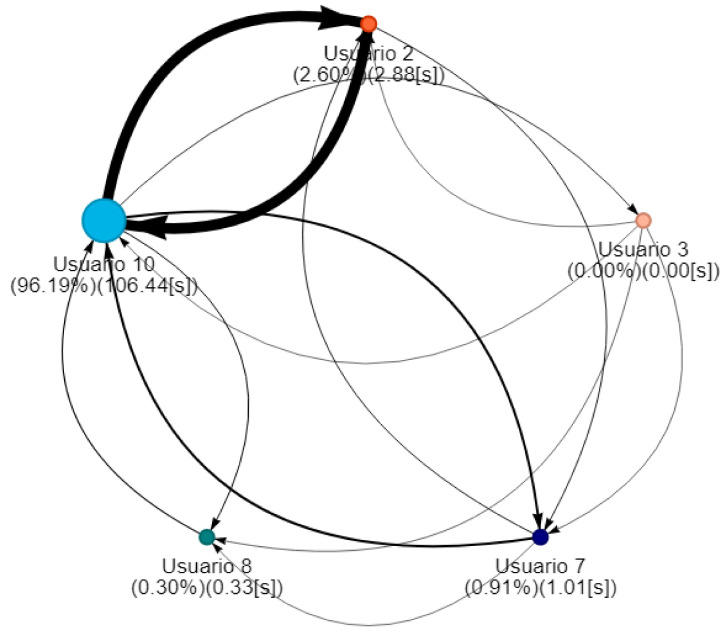
Problem detection with G5.

**Figure 9 sensors-20-06337-f009:**
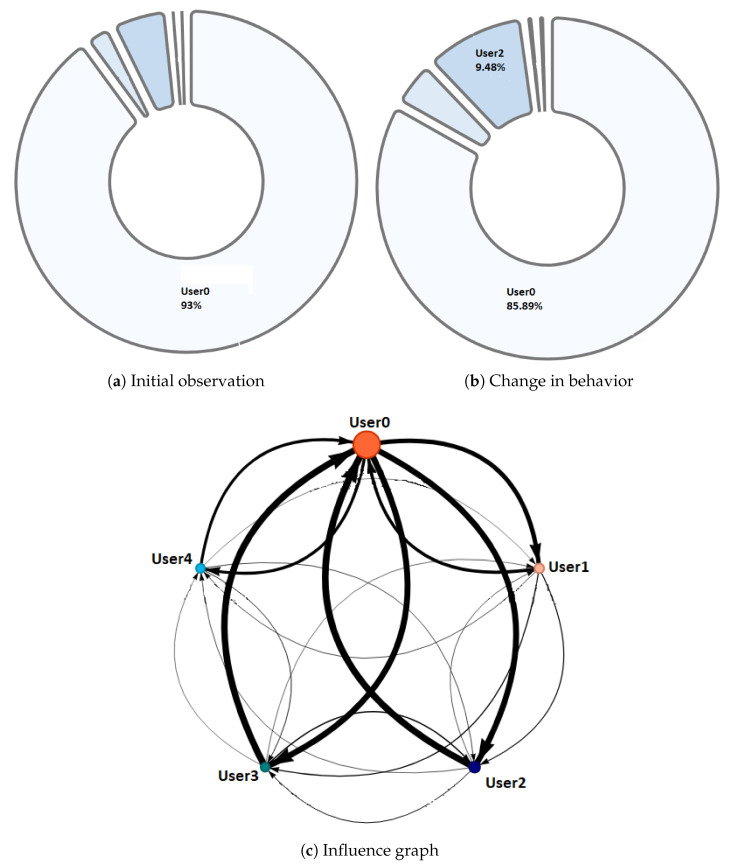
Behavior observed in G3 during phase 2.

**Figure 10 sensors-20-06337-f010:**
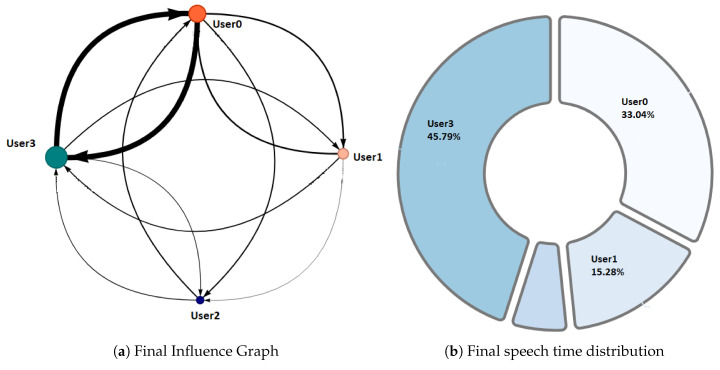
Final Behavior observed in G2.

**Figure 11 sensors-20-06337-f011:**
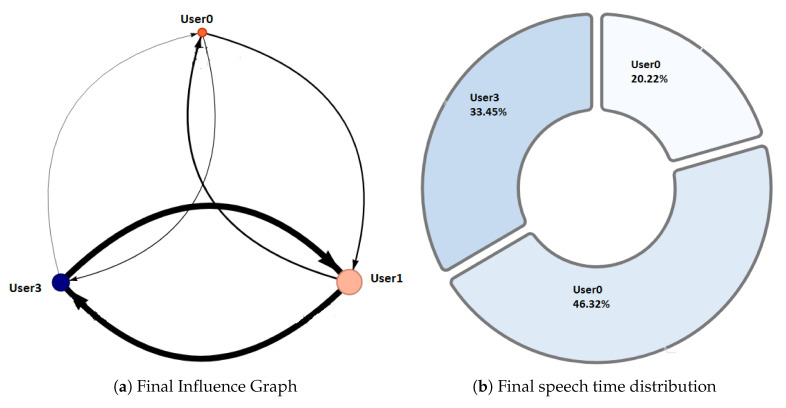
Final Behavior observed in G4.

**Figure 12 sensors-20-06337-f012:**
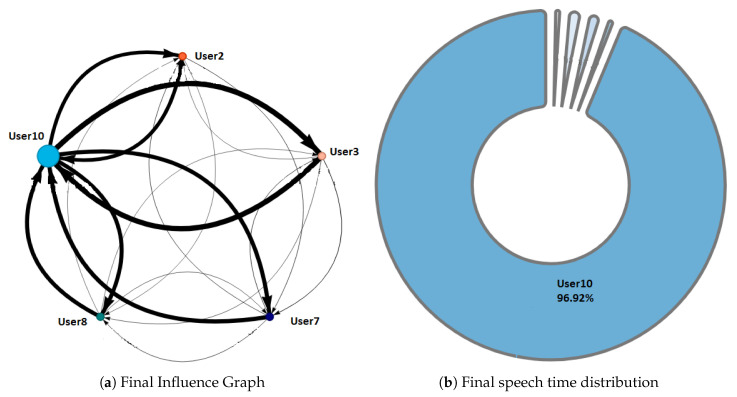
Final Behavior observed in G5.

**Figure 13 sensors-20-06337-f013:**
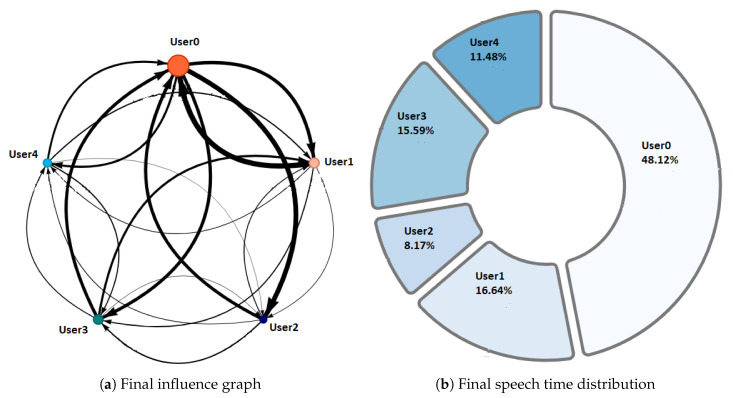
Final Behavior observed in G1.

**Figure 14 sensors-20-06337-f014:**
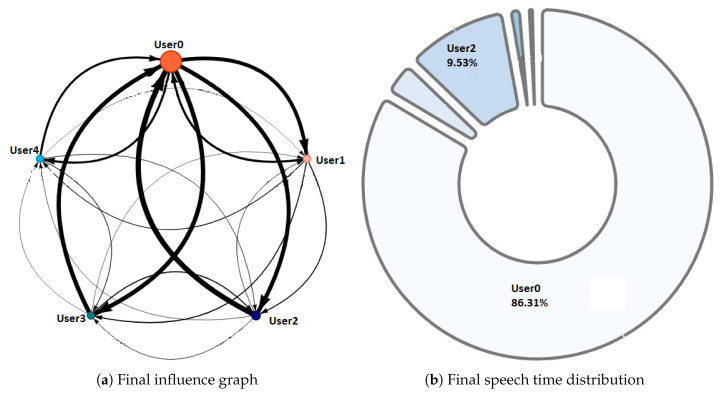
Final behavior observed in G3.

**Figure 15 sensors-20-06337-f015:**
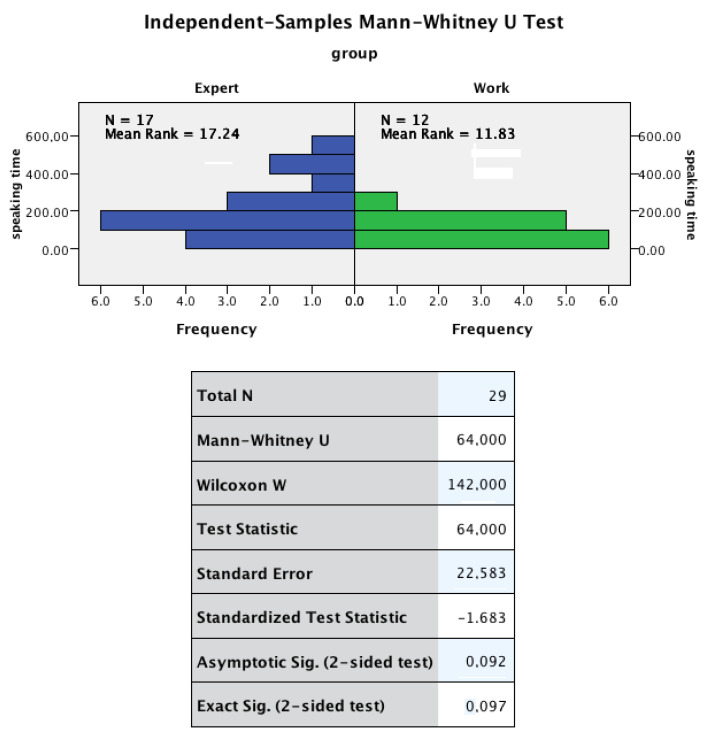
Independent-Samples Mann–Whitney U Test for student speaking time among expert and work groups.

**Figure 16 sensors-20-06337-f016:**
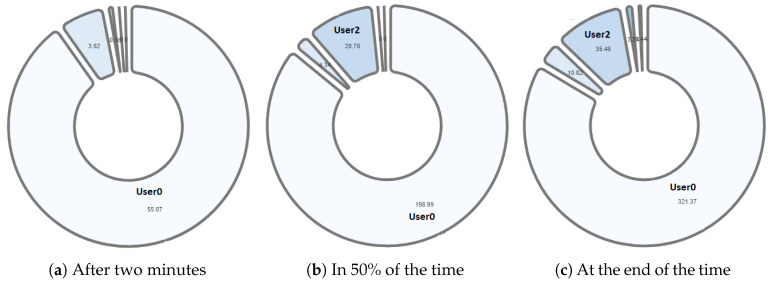
Pie chart with the evolution of G3 speech time distribution.

**Figure 17 sensors-20-06337-f017:**
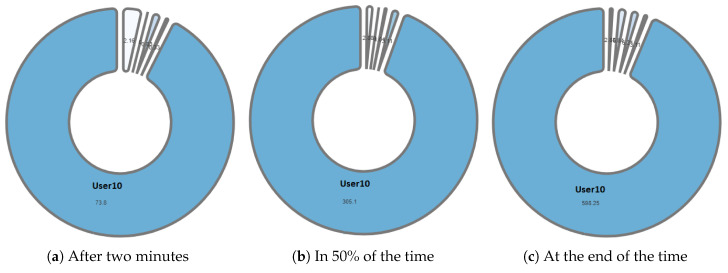
Pie chart with the evolution of G5 speech time distribution.

**Figure 18 sensors-20-06337-f018:**
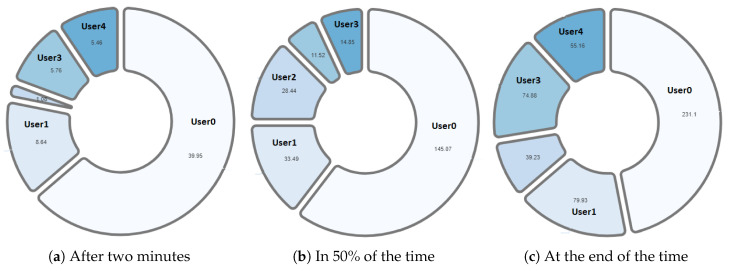
Pie chart with the evolution of G1 speech time distribution.

**Figure 19 sensors-20-06337-f019:**
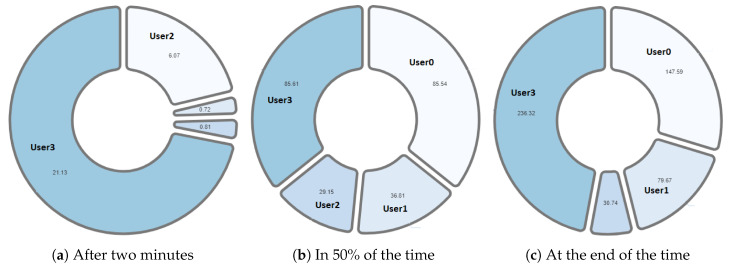
Pie chart with the evolution of G2 speech time distribution.

**Figure 20 sensors-20-06337-f020:**
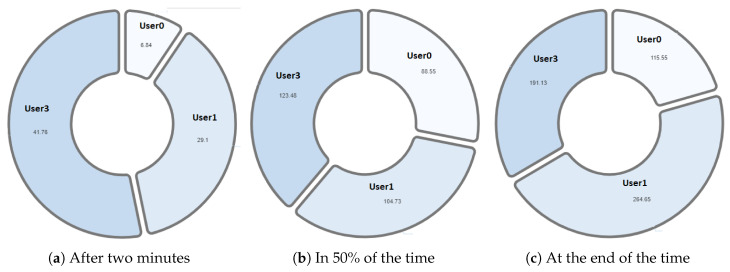
Pie chart with the evolution of G4 speech time distribution.

**Table 1 sensors-20-06337-t001:** Number and duration of events registered by the teacher and observers.

Event	n	Duration	Average	Sd	Max	Min
Teacher’s observation	5	277	55	20	87	35
Teacher’s intervention	7	307	44	25	96	20
Group observers	8	518	65	32	111	21

**Table 2 sensors-20-06337-t002:** Number and duration of events by group.

Event	G1		G2		G3		G4		G5	
	n	Time	n	Time	n	Time	n	Time	n	Time
Teacher’s observation	2	112	2	98	3	199	2	98	2	90
Teacher’s intervention	1	20	2	86	1	25	1	40	2	136
Group observers	3	147	4	253	3	215	5	297	4	280
Total	6	279	8	437	7	439	8	435	8	506

**Table 3 sensors-20-06337-t003:** Survey results of group communication, group members’ contribution, and activity design.

Survey Item	Median	Mode
Group Communication		
GC1	5	5
GC2	1	1
GC3	1	1
Group Members Contribution		
MC1	5	5
MC2	5	5
MC3	5	5
MC4	5	5
MC5	5	5
MC6	4	5
MC7	5	5
Activity Design		
AD1	5	5
AD2	5	5
AD3	5	5

**Table 4 sensors-20-06337-t004:** Data collected in the case study. id: student id, exp grp: expert group, st(exp): speaking time in the expert group stage, ni(exp): number of interventions in the expert groups stage, wrk grp: work group, st(wrk): speaking time in the work groups stage, ni(wrk): number of interventions in the work groups stage.

id	exp grp	st(exp)	ni(exp)	wrk grp	st(wrk)	ni(wrk)
1	1	401	592.90	1	835	179.32
2	2	136	30.00	1	145	15.20
3	3	425	28.80	1		
4	4	298	135.00	1	322	18.70
5	5	968	373.40	1	420	38.90
6	1	24	7.00	2	79	29.40
7	2	185	130.00	2	425	115.80
8	3	143	79.90	2		
9	4	183	155.70	2	354	189.50
10	1	277	206.30	3		
11	2	224	131.40	3		
12	3			3		
13	4	379	223.60	3		
14	5	683	217.40	3		
15	1	230	163.10	4	165	156.60
16	2	365	455.10	4	193	270.30
17	3			4		
18	4	344	419.70	4	72	105.50
19	5	353	90.20	4		
20	1			5		
21	2	16	10.70	5		
22	3			5		
23	4			5		
24	5	510	134.70	5

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
