# Peer review of "A Multimodal Real-Time Feedback Platform Based on Spoken Interactions for Remote Active Learning Support"

_sensors, 2020, doi:10.3390/s20216337_

Round 1

Reviewer 1 Report

In this paper, a platform to help teachers to monitor working groups is presented. The platform with the different components is described and a case study is analysed. The results are promising, because the real-time feedback provided by the proposed system was highly valuable for the teacher to take pedagogical actions.

I congratulate the authors because I think this proposal is quite interesting and helpful for teaching-learning processes. The following comments aim to help to improve the quality of this work.

I have a concern about the title of this work, because it says "based on lavalier microphones", but later in the manuscript, the authors write:

155 NAIRA App is a multi-platform mobile app (Android, iOS) that allows capturing spoken
156 interaction data through the mobile device microphone, or lapel microphone connected to the
157 smartphone, for more accurate results

Therefore, I think "based on lavalier microphones" should be removed from the title, because it is not only "based on" that kind of microphones.

I have another concern about the technology used in the system. For example, why Firebase Cloud Platform? Did the authors evaluate other options?

Related to this, the data is processed by cloud services, Google Cloud Platform as stated in section 3.4. Cloud Back-End Services. What is the cost of these services?

Another concern is the development of the NAIRA app? Why native mobile applications? Why not a hybrid application developed in, for example, Ionic? Did you analyse the benefits and drawbacks?

One suggestion: I think it could be interesting to correlate the participation of the students (according to the NAIRA system) and their responses to the survey after the activity. This analysis could provide some interesting insight in their participation.

Some specific comments:

In the abstract, the acronym "MMLA" should be defined. In addition, in the sentence:

11 to verify that the components of the platform provided the teacher with tools that helped him to
12 facilitate remote learning activity in an optimal way.

The pronoun "him" should be changed to avoid gender bias. Current style guidelines encourage "Bias-Free Language", for example:

https://apastyle.apa.org/style-grammar-guidelines/bias-free-language/general-principles

https://apastyle.apa.org/style-grammar-guidelines/bias-free-language/gender

Some statements require references to support them. For example:

16 Technological development in the first decades of the 21st century has grown exponentially.

How do the authors know that the growth has been exponentially?

The text in some figures is quite small and it is difficult to read. For example: Figure 2, 8-13.

In Figure 2, some content is in Spanish.

Figure 2 and Figure 8 have the same caption: "Behavior observed in G3 during phase2.". Each figure should have a unique caption.

In section 4.1. Case Study Definition, the Jigsaw Learning Activity Design is presented for the first time. Although later in the manuscript the Jigsaw is describes, I miss a brief explanation in this part of the text because not all reader will know it.

In Figure 5, the labels say "Professors", but in the rest of the manuscript the term "Teacher" is used.

In line 336 the authors say:

The story was complemented with the teacher field notes, which also have the timestamp of the events noted.

How were the timestamps generated? By hand? Automatically? What is the precision?

Regarding the references, some references must be completed or improved. As an example:

10. JAIN, A.; DwIvEDI, P.K. The Evidence for the Effectiveness of Active Learning 2014.

The correct reference is:

Anurag Jain A. J, Dwivedi P. K. The Evidence for The Effectiveness of Active Learning. Orient.J. Comp. Sci. and Technol;7(3)

Or in:

26. Praharaj, S.; Scheffel, M.; Drachsler, H.; Specht, M. Group Coach for Co-located Collaboration. European Conference on Technology Enhanced Learning. Springer, 2019, pp. 732–736.

Springer should be removed, because the publisher does not appear in other references.

Some minor typos:

37 "black Screens" --> Lower case, "screens".

Figure 2 caption: Behavior observed in G3 during phase2 --> "phase 2".

Finally, although this is not the topic of this work, I am very interested in this information:

37 Teaching to “black Screens”, thus, to students that do not turn on their cameras during
38 a class, is not just a challenge for the social interaction among the teacher and the students but to
39
effectively assess if students are learning, as well as to implement different active learning strategies.

Could you include any reference about analysis of the "black screen" problem? Solutions?

Author Response

Dear Reviewer1

Best regards!

Reviewer 2 Report

The abstract does not present any research questions, hypothesis, methodology, results, or implications.

I find the paper too descriptive, it looks more like a product presentation, not like a research paper. I would appreciate it if the authors conducted research into the efficiency of the app and NAIRA web.

Even the part called empirical evaluation does not contain relevant data and their presentation so that the reader could evaluate the scientific soundness of the case study.

The results are not interpreted and this section is very short. I guess this should be the most important part of the paper.

The discussion does not contain any discussion, i.e., there is no relevant research to be compared with the manuscript data. I would expect further implications for the practical utilization of the platform with a potential future outlook for the implementation of similar devices in education.

The figures are in Spanish and very difficult to read, therefore, almost impossible to evaluate their soundness and relevance. There is not much interpretation of the data obtained and presented in the figures either.

I find the findings, i.e. “this experience allowed us to demonstrate the importance of the use of technologies to support academic activities today “ as not sufficient for an impact paper. It would be necessary to clearly demonstrate, based on wide-scale research, the benefits and the drawbacks of the given platform, which I cannot see in the manuscript at all.

Author Response

Dear Reviewer2

Best regards!

Round 2

Reviewer 1 Report

I have reviewed the author's response to the previous revision, I have reviewed the new version of the manuscript and I think the authors have implemented all my suggestions. The quality and content of the article has improved substantially, and I congratulate the authors because they have done a good revision of their manuscript. Therefore, I do not have more objections and I think this paper could be accepted.

I have some comments to improve the quality of the manuscript:

+ The size and quality of the figures should be revised and improved. For example, if top of Figure 2 and 9 (10, 11, 12, 13) are compared, they are the same type of figure, but they do not have the same size. And in Figure 9, the size of the text is smaller and it is difficult to read.

+ Regarding Figure 2, it has different parts (a, b, c). I think it is better if the figure is divided into parts: a + b and c.

+ The dot/period (.) should be used as decimal separator in the whole document. For example, in the table in Figure 2 is correctly used, but not in Table 4 or in Figure 14. The dot/period is the preferred separator in English.

Reviewer 2 Report

The authors have implemented all my suggestions and I feel the paper has been improved by their effort so that now it is clearer and more systematic. Therefore, there is nothing not to warrant the publication of the paper if the academic editor considers it relevant for the issue and significant for the academic discussion.

On the part of the reviewer, there is not much to object with. The authors have followed standard methodology with relevant findings. Of course, there are many other ways how to improve the manuscript, but they are not necessary to be implemented.

I would just recommend a professional proofreading service so that the paper is perfect from the language point of view too. 
